# Understanding the Anti-Diarrhoeal Properties of Incomptines A and B: Antibacterial Activity against *Vibrio cholerae* and Its Enterotoxin Inhibition

**DOI:** 10.3390/ph15020196

**Published:** 2022-02-03

**Authors:** Fernando Calzada, Elihu Bautista, Sergio Hidalgo-Figueroa, Normand García-Hernández, Claudia Velázquez, Elizabeth Barbosa, Miguel Valdes, Jesús Iván Solares-Pascasio

**Affiliations:** 1Unidad de Investigación Médica en Farmacología, UMAE Hospital de Especialidades, 2° Piso CORSE, Centro Médico Nacional Siglo XXI, Instituto Mexicano del Seguro Social, Av. Cuauhtémoc 330, Col. Doctores, Cd Mexico 06725, Mexico; valdesguevaramiguel@gmail.com (M.V.); jeivsp@gmail.com (J.I.S.-P.); 2CONACYT-División de Biología Molecular, Instituto Potosino de Investigación Científica y Tecnológica A.C., San Luis Potosí 78216, Mexico; francisco.bautista@ipicyt.edu.mx (E.B.); sergio.hidalgo@ipicyt.edu.mx (S.H.-F.); 3Unidad de Investigación Médica en Genética Humana, UMAE Hospital Pediatría 2º Piso, Centro Médico Nacional Siglo XXI, Instituto Mexicano del Seguro Social, Av. Cuauhtémoc 330, Col. Doctores, Cd Mexico 06725, Mexico; normandgarcia@gmail.com; 4Área Académica de Farmacia, Instituto de Ciencias de la Salud, Universidad Autónoma del Estado de Hidalgo, Km 4.5, Carretera Pachuca-Tulancingo, Unidad Universitaria, Pachuca 42076, Mexico; cvg09@yahoo.com; 5Sección de Estudios de Posgrado e Investigación, Escuela Superior de Medicina, Instituto Politécnico Nacional, Salvador Díaz Mirón Esq. Plan de San Luis S/N, Miguel Hidalgo, Casco de Santo Tomas, Cd Mexico 11340, Mexico; rebc78@yahoo.com.mx

**Keywords:** incomptine A, incomptine B, sesquiterpene lactones, *Decachaeta incompta*, *Vibrio cholerae*, cholera toxin, cholera toxin-induced diarrhoea, SDS-PAGE analysis, docking analysis

## Abstract

Incomptines A (**IA**) and B (**IB**) are two sesquiterpene lactones with antiprotozoal, antibacterial, cytotoxic, antitumor, spermicidal, and phytotoxic properties. The antibacterial activity of **IA** and **IB** against bacteria causing diarrhoea have been reported; however, no information is available regarding their antibacterial activity on *Vibrio cholerae*. In this work, both compounds were evaluated for their anti-diarrhoeal potential using the bacterium *V. cholerae*, sodium dodecyl sulphate–polyacrylamide gel electrophoresis (SDS-PAGE) analysis on cholera toxin, and a cholera toxin-induced diarrhoea model in male Balb/c mice. In addition, a molecular docking study was carried out to understand the interaction of **IA** and **IB** with cholera toxin. In terms of antibacterial activity, **IB** was three times more active than **IA** on *V. cholerae*. In the case of SDS-PAGE analysis and the in silico study, **IA** was most effective, revealing its potential binding mode at a molecular level. In terms of anti-diarrhoeal activity, **IA** was 10 times more active than **IB** and racecadotril, an antisecretory drug used as positive control; the anti-diarrheal activity of **IB** was also closer than racecadotril. The results obtained from in vitro, in vivo, and computational studies on *V. cholerae* and cholera toxin support the potential of **IA** and **IB** as new anti-diarrhoeal compounds.

## 1. Introduction

Worldwide gastrointestinal infections are the most common cause of diarrhoea, and are an important cause of morbidity and mortality in developing countries. Diarrhoeal diseases (DDs) are causes of millions of deaths every year, affecting all age groups, but mainly children younger than 5 years [1,2,3,4]. In Mexico, DDs are an important health problem, and are the second greatest cause of morbidity among groups of all ages [5]. There are a wide range of pathogens that can cause DD, including enterotoxin-producing bacteria strains such as *Escherichia coli*, *Clostridium difficile*, *Salmonella typhi*, *Yersinia enterocolitica*, and *Vibrio cholerae*. Among these, *V. cholerae* causes an acute infectious diarrhea known as cholera, which occurs when the balance between absorption and secretion in the small intestine is altered by high secretion provoked by its enterotoxin. Heat-labile enterotoxin from *V. cholerae* induces the intestinal hypersecretion of fluid and electrolytes by activation of the cyclic AMP-adenylate cyclase system in the mucosal epithelium of the small intestine [1,2,3]. Cholera has been catalogued as an emergent or re-emergent disease that threatens developing countries. This bacterium has a short incubation period and, in consequence, cases can rise extremely rapidly, with nearly 10–20% of patients developing severe watery diarrhoea and vomiting. This leads to dehydration, which can result in hypotensive shock, renal failure, and death within hours of onset. Cholera also affects all age groups; nevertheless, deaths caused by cholera are most frequent in children younger than 5 years. It is estimated that there are between 1.4 and 4 million cases of cholera per year, and estimated that cholera deaths range from 21,000 to 143,000. The spread form consists in the fecal–oral route, either from person-to-person or through contaminated fluids, food, water, and fomites. Likewise, outbreaks can occur where water supply, food safety, and hygiene are inadequate [6,7]. Oral or intravenous rehydration therapy is regarded as the treatment of choice to control cholera. It reduces the levels of mortality from dehydration, but does not reduce the morbidity of cholera [6]. For the treatment of acute diarrhoea, there are some drugs, such as racecadotril, loperamide, and bismuth salicylate, that decrease intestinal hypersecretion. However, these drugs have side effects including fever, bronchospasm, vomiting, tinnitus, constipation, black tongue, and intestinal obstruction [1]. Thus, the search for anti-diarrhoeal agents that are effective and safe for the treatment of cholera is still a necessary goal.

The Asteraceae family is a group of plants whose members are rich in phytochemicals such as sesquiterpene lactones (SLs). These secondary metabolites are found mainly in genus *Decachaeta*, *Artemisia*, *Ambrosia*, *Helenium*, *Tanacetum*, and *Vernonia*, which show a broad range of biological activities including antimicrobial, antiviral, antifungal, anti-inflammatory, cytotoxic, and anti-tumor effects [8,9]. *Decachaeta* genus contains eight species, including *Decachaeta incompta*, *D. perornata*, *D. haenkeana*, *D. pyramidalis*, *D. ovatifolia*, *D. scabrella*, *D. ovandensis*, and *D. thieleana,* which are endemics of Mexico to Central America [10]. In this region, some *Decachaeta* species are used in traditional medicine to treat gastrointestinal disorders such as diarrhoea and stomach aches [11]. Of the eight species, *D. incompta* (DC) King and Robinson is the most studied; it is a herbaceous plant from which four heliangolide-type SLs, incomptines A-D, have been obtained and evaluated for their spermicidal, allelopathic, antibacterial, antiprotozoal, antipropulsive, cytotoxic, and antitumor properties, as well as for their lethality on *Artemia salina*. Bioassay-guided fractionation of the dichloromethane extract from the leaves of *D. incompta* using antiprotozoal and antibacterial assays showed incomptine A (**IA**) as the most active antiprotozoal compound on *Entamoeba histolytica* and *Giardia lamblia*, and incomptine B (**IB**) as the most effective antibacterial sesquiterpene lactone on Gram-negative chloramphenicol-resistant bacteria strains, including, *Escherichia coli*, *Shigella sonnei*, and *S. flexneri* [12,13]. In this sense, this strong antiprotozoal activity may be associated with the presence of 8-acetate moiety of the germacrene framework. In contrast, this antibacterial activity is apparently related to 8-OH moiety [12,13]. In addition, both SLs have in their structure a 3-methylenedihydrofuran-2-(3*H*)-one moiety, which is associated with their potent bioactivities, including phytotoxic, cytotoxic, antitumor, antiulcer, and schistosomicidal effects [8,9,14,15].

This background prompted us to evaluate the antibacterial activity on *Vibrio cholerae*, and the in vivo, in vitro, and in silico effects of incomptines A (**IA**) and B (**IB**) (Figure 1), on cholera toxins. The incomptines were obtained from the dichloromethane extract of the leaves of *D. incompta* as a part of our search for new anti-diarrhoeal natural products.

## 2. Results

### 2.1. Antibacterial Activity, Cholera Toxin-Induced Diarrhoea, and SDS-PAGE Analysis on Cholera Toxin of Sesquiterpene Lactones from Decachaeta incompta

The antibacterial activity against the bacterium *Vibrio cholerae* showed (Table 1) that the sesquiterpene lactone incomptine B (**IB**), with a minimal inhibitory concentration (MIC) value of 0.05 mg/mL, was three time more active than incomptine A (**IA**). The antibacterial activity of both SLs was superior to that of chloramphenicol (**CLO**), an antibiotic drug used as a positive control, which had an MIC value > 2 mg/mL. In addition, anti-diarrhoeal activity of the **IA** and **IB** were assayed on a cholera toxin-induced diarrhoea model in male Balb-c mice. The results showed (Table 1) that **IA** had an inhibitory effect with an ED_50_ of 8.1 mg/kg; its effect was 10-fold more active than **IB** and racecadotril, an antisecretory drug used as the control. **IB** showed anti-diarrhoeal activity close to that of racecadotril.

### 2.2. Sodium Dodecyl Sulphate-Polyacrylamide Gel Electrophoresis (SDS-PAGE) Analysis on Cholera Toxin of Incomptine A (IA) and Incomptine B (IB)

To characterize the mode of action of incomptine A (**IA**) and incomptine B (**IB**) on cholera toxin, we analyzed the interaction between different concentrations of SLs and cholera toxin using SDS-PAGE analysis (Figure 2 and Figure 3). The results revealed that **IA** interacted with cholera toxin at concentrations from 0.64 mg to 2.56 mg, and its effect was greater on the subunit B than subunit A. In contrast, **IB** was inactive at the concentrations tested.

### 2.3. Molecular Docking Studies of Incomptine A (IA), Incomptine B (IB), and Racecadotril (REC) on Cholera Toxin

Based on previous in vitro results, we decided to study three potential ligandable sites from cholera toxins [16,17]. The results of residues of interaction between incomptine A (**IA**), incomptine B (**IB**), and racecadotrile (**REC**; *R*/*S*-enantiomers) and three considered binding pockets (Figure 4 and Table 2) showed that **IA** was the most potent compound, even more potent than racecadotril. Through the prediction studies of the binding modes on the ligandable sites, we can observe that **IA** had a greater affinity on the two sites of the subunit B with ΔG, −6.6 kcal/mol (GM1) and −6.5 kcal/mol (HBGA), respectively; its affinity was greater than **IB**. Moreover, its affinity on the secondary binding site (HBGA) was greater than (±)-**REC.** In the case of the catalytic site in the subunit A with ΔG −5.9 kcal/mol, its affinity was less than **IB**. In addition, **IA** had one hydrogen bond with three binding pockets. In the subunit B, Gly33 was at first binding site and Lys84 was at the second binding site of CTB; in the subunit A, Gln49 was at the binding site of CTA. In the case of **IB,** it showed less affinity on the two sites of the subunit B with ΔG, −6.1 kcal/mol and −6.1 kcal/mol, respectively. In contrast, its affinity for the catalytic site in the subunit A was greater than that of **IA** and (±)-**REC**. In addition, this lost the two interactions by hydrogen bonds in the two sites of the subunit B. Concerning racecadotril, each enantiomer showed similar affinity compared with each other in the mentioned sites. They also showed the best affinity for primary binding sites (GM1) in the subunit B, rather than **IA** and **IB**.

Incomptine A only has two rotatable bonds, which are underlined in bold (Figure 5). We analyzed the torsion angle values of rigid ligands from molecular docking experiments, compared with the structure with the lowest energy obtained by conformation analyses.

The most stable conformation was obtained with a calculated energy value of 34 kcal/mol. Then, the dihedral (or torsion) angles were measured between the structure of incomptine A (lowest energy structure from conformational analysis) and each pose of the site A and B obtained from the molecular docking study. The torsion angle defined by the sequentially bonded atoms represented (Figure 5) as C1-O1-C2-O2 were taken into account for this purpose, and these were compared with other molecules from the molecular docking study. We found that the carbonyl group (O2) of an ester moiety with carbon atom C1 had a torsion angle of 1.1° (eclipsed conformation). The same O2 of the pose from **IA** into the active site (CTA), with respect to C1, had a torsion angle of −154.8° (staggered conformation). The structure obtained into site B (first binding site CTB) had a torsion angle of 124.8° (staggered conformation) and the pose into site B (second binding site CTB) had a torsion angle of 36.3° (eclipsed conformation). All representations are illustrated in Figure 6.

Finally, we can observe that the bond conformation of **IA** into site B (second binding site CTB) had the closest torsion angle to the lowest energy conformer. This particular orientation of the carbonyl group led to generating a high affinity to the second binding site of CTB. However, both staggered and eclipsed conformations induced great affinity inside the two allosteric sites, with similar binding energy between them. Due to the fact that the structure of **IA** had a few variations in its structural conformation, these small changes in its orientation gave it a similar affinity towards the 2 allosteric sites, although it lost affinity towards the active site of the cholera toxin, which may explain the high antagonistic activity of **IA**.

## 3. Discussion

Leaves of *Decachaeta incompta* have been used in Oaxaca, México, for the treatment of some gastrointestinal disorders such as diarrhoea [12]. Other *Decachaeta* species have been well documented for their anti-diarrhoeal activity, supporting their traditional uses [11]. We are undertaking a continued search for potential anti-diarrhoeal compounds from plants commonly used in Mexican traditional medicine for the treatment of gastrointestinal disorders such diarrhoea, dysentery, and abdominal pain [18]. In this work, two SLs, incomptines A (IA) and B (IB), isolated of the leaves from *Decachaeta incompta* were evaluated for their anti-diarrhoeal potential using in vivo, in vitro, and in silico experiments including antibacterial activity against *Vibrio cholerae*, sodium dodecyl sulphate-polyacrylamide gel electrophoresis (SDS-PAGE) analysis on cholera toxin, anti-diarrhoeal activity on cholera toxin-induced diarrhoea models in male Balb/c mice, and by molecular docking on the cholera toxin. The results obtained (Table 1) showed that incomptine A (IA) and incomptine B (IB) exhibited antibacterial activity on the bacterium *Vibrio cholerae* with minimal inhibitory concentration (MIC) values of 0.150 mg/mL and 0.05 mg/mL, respectively. Their antibacterial activity was superior to chloramphenicol, the antibiotic drug used as the control. Since IB was three times more active than IA, the antibacterial activity on *V. cholerae* seemed to be related to the C-8 substituent, suggesting that the 8-hydroxy group of the germacrene ring is important for the significant antibacterial effect. Our result is in agreement with the previous report that IB is more active than IA on six Gram-negative chloramphenicol-resistant bacteria strains, including *Escherichia coli*, *Shigella sonnei*, and *S. flexneri* [12]. In this context, it is noteworthy that the *V. cholerae* bacterium used was resistant to chloramphenicol. This result indicates that IB may be a potential antibacterial agent on chloramphenicol-resistant bacteria strains in particular, including *V. cholerae*. Moroever, this result, along with the antipropulsive and antiprotozoal activity previously reported [12,18] from the dichloromethane extract of the leaves of *Decachaeta incompta,* and its major SLs, incomptines A (A) and B (IB), could suggest that the mechanism by which the SLs and the plant inhibit the diarrhoea involves antibacterial, antiprotozoal, and antipropulsive effects.

Once the antibacterial activity of the IA and IB on *V. cholerae* were demonstrated, we decided to evaluate their activity as anti-diarrhoeal agents, in particular on a cholera toxin-induced diarrhoea model in male Balb/c mice. Thus, evaluating the effects of IA and IB on male Balb-c mice with diarrhoea induced by cholera toxins with a concentration of 10 μg/mL x mouse showed significant anti-diarrhoeal activity with an ED_50_ value of 8.1 mg/kg for IA. The anti-diarrhoeal activity of IA was in agreement with the previously reported data for the flavan-3-ol, (-)-epicatechin (ED_50_ 14.7 mg/kg) with significant anti-diarrhoeal activity. The anti-diarrhoeal activity of IA was 10-fold more active than IB and racecadotril (REC), an antisecretory drug used as the positive control [1]. These data give additional support that IA may play an important role in the anti-diarrhoeal properties of the *D. incompta*. The above support that IA may be used as a potential phytochemical for the development of novel anti-diarrhoeal drugs, specifically against acute secretory diarrhoea [18,19,20]. In addition, the different significant properties suggest that IA and IB may have complementary anti-diarrhoeal effects in the leaves of *D. incompta*, the first as an antisecretory agent and the second as an antibacterial compound.

To understand how the SLs interacted with cholera toxin, we evaluated the effect between the different concentrations of IA and IB and the toxin, using the SDS-PAGE assay (Figure 2). The results showed that IA showed a different inhibition on the subunits A and B from the cholera toxin at a concentration from 0.64 mg to 2.56 mg. Subunit A of the cholera toxin (CT) was inhibited 43% at 2.56 mg of concentration. In contrast, the subunit B of CT was inhibited 100% at a concentration of 1.28 mg of IA. The SL, incomptine B (IB) was inactive at the doses tested in this model. This observation indicates that the mode of toxin inhibition depends upon the concentration of applied SL. A similar effect was reported to resveratrol. Moreover, it could be that incomptines A and B caused the aggregation of the cholera toxin. Additional studies are necessary to discard the potential effect of IB on cholera toxin [6]. The structure-effect correlation revealed that the 8-acetate group in the germacrene framework may be important to the inhibition effects on the cholera toxin, as occurred with the antiprotozoal effect [13]. In this sense, a similar result was reported to phytotoxic activity, where substituent in C-8 was important to obtain significant activity [21].

Docking analysis showed that IA had a strong affinity for the two binding sites in the subunit B present in the cholera toxin, and its effect was more significant than that of incomptine B (IB) and racecadotril, an antisecretory drug used as positive control. These results are in agreement with the results from the cholera toxin-induced diarrhoea model in male Balb/c mice and the SDS-PAGE assay. Moreover, we suggest that IA may be a potential agent with utility for the treatment of acute secretory diarrhoea, in particular that caused by *Vibrio cholerae*. In contrast, IB in the docking analysis showed a greater affinity on the catalytically active site in the subunit A, than IA. The results of docking analysis along with the SDS-PAGE assay suggested that IA and IB may have a dual effect on the cholera toxin as a mechanism of action. Firstly, IA and IB may prevent toxin binding at the cell surface in the gut, by the disruption of the subunit B interaction with its GM1 receptor, in consequence inhibiting toxin internalization and toxin activity; secondly, IA and IB appeared to directly inhibit the catalytic activity of subunit A on the cholera toxin. To our knowledge, this is the first report of the antibacterial activity against the bacterium *V. cholerae* and the cholera toxins of SLs, incomptines A (IA) and B (IB).

## 4. Materials and Methods

### 4.1. Collection and Identification of Decachaeta incompta

The aerial parts of *Decachaeta incompta* (D.C.) King and Robinson (Asteraceae) were collected in Portillo Nejapa de Madero (16°36′00″ N, 95°59′00″ O) State of Oaxaca, Mexico. The plant material was authenticated by MSc Abigail Aguilar and a voucher specimen (voucher 15311) was deposited at the medicinal Herbarium IMSSM of the Instituto Mexicano del Seguro Social (IMSS).

### 4.2. Animals

Male Balb-c mice (25–30 g) were obtained from the animal house of the IMSS. These studies were conducted with approval of the Bio-Ethical and National Scientifical Research Committees of the National Medical Center Siglo XXI from IMSS (Approval No: R-2018-785-111). The investigation using experimental animals was conducted in accordance with the official Mexican norm NOM 0062-ZOO-1999, [22] with technical specifications for the production, care, and use of laboratory animals. Animals were maintained with a 12 h light–dark cycle at 22 °C ± 2 °C at the controlled condition. They were fasted overnight, but tap water was available ad libitum until the start of the experiment.

### 4.3. Chemicals

Racecadotril was purchased from Ferrer Therapeutics, S.A. de C. V. (Hidrasec, *Rac*-racecadotril; capsule of 100 mg); dimethyl sulfoxide (DMSO), cholera toxin from *Vibrio cholerae* (>90 %, SDS-PAGE, lyophilized powder, C8052), Coomassie-blue (B0149), Muller-Hinton agar (70191), chloramphenicol (C0378), acetonitrile HPLC grade, acetic acid HPLC grade were purchased from Sigma-Aldrich, St. Louis, MO, USA. EtOH, dichloromethane, hexane, and MeOH AR grade were purchased from JT Baker, Mexico City, Mexico.

### 4.4. Isolation of Sesquiterpene Lactones of the Aerial Parts from Decachaeta incompta

The air-dried plant material (25 g) were ground and extracted by percolation at room temperature with dichloromethane (350 mL). After filtration, the solvent was evaporated *in vacuo* to yield 2 g of brown residue. The dichloromethane extract (1.8 g) was subjected to column chromatography (CC) over Silica gel (20 g, 70–230 mesh, Merck), using hexane and a mixture of dichloromethane-MeOH (7:3–5:5) to give five fractions (Fr_1_–Fr_5_). Fractions 3 and 4 were combined and resolved by CC over silica gel (20 g), using a mixture of solvents and dichloromethane in MeOH (7:3–5:5) to yield **IA** (95 mg). Fr_5_ was purified by crystallization with dichloromethane-hexane (7:3) to obtain **IB** (600 mg). Incomptines A (**IA**) and B (**IB**) were identified by comparison thorugh ^1^H-NMR, IR and HPLC-DAD with authentic sample disposables in our laboratory. Incomptine A: white crystals; mp 175–176 °C. Incomptine B: white crystals; mp 178–179 °C [12].

### 4.5. Antibacterial Assay

The samples (**IA**, **IB**, and chloramphenicol) were tested on the bacterium *Vibrio cholerae*. The bacterial inoculum of *V. cholerae* was obtained from fresh colonies grown on Muller–Hinton agar plates (MHA, Sigma). The bacterial strain used in this study was isolated from the faeces of children with acute diarrhoea [23]. Moreover, the strain tested was resistant to chloramphenicol. The determination of minimum inhibitory concentration (MIC) of samples was accurately determined by the agar dilution technique [24]. Briefly, pure compounds for testing were dissolved in DMSO, and serially diluted in melted MHA plates (100 mm × 15 mm) to obtain the final concentrations of 50, 100, 150, 200, and 2000 μg/mL. The solvent did not exceed 1 % concentration and did not affect the growth of the microorganisms. The culture was diluted in Muller–Hinton broth at a density adjusted to a 0.5 McFarland turbidity standard (1.5 × 10^8^ colony-forming units (CFU)/mL). Then, the inoculum was added to a Steer’s replicator calibrated to deliver 10^4^ CFU. After this, Petri dishes were inoculated and incubated at 37 °C, examined after 24 h, and further incubated for 72 h. Chloramphenicol was used as reference standard and for comparative purposes. The lowest concentration of the sample in a plate that failed to show any visible macroscopic bacterial growth was considered as the MIC. The MIC determination was performed in duplicate for each organism, and the experiment was repeated two times.

### 4.6. Cholera Toxin-Induced Diarrhoea

Mice were randomly allocated in groups of six mice per group, with free access to water. Diarrhoea was induced in the experimental groups by the cholera toxin (0.5 mL of cholera toxin at a concentration of 10 μg/mL × mouse) except for a blank. Thirty minutes after, materials were administrated as follows: blank (0.5 mL of 2 % DMSO solution in water), **IA**, **IB**, and rececadotril (**REC**) (1, 10, 20, 40, 80, and 120 mg/kg in 0.5 mL of a 2 % DMSO in water). All test materials and cholera toxins were administered intragastrically. Immediately after administration, the animals were placed in cages lined with adsorbent paper and were observed for 4 h; then, the total mass of fecal output (mg) was measured and expressed in the percentage of inhibition.

### 4.7. Sodium Dodecyl Sulphate-Polyacrylamide Gel Electrophoresis (SDS-PAGE)

*Vibrio cholerae* toxin (10 μg) dissolved in water was treated with incomptine A (**IA**) or incomptine B (**IB**) (0, 0.64 mg, 1.28 mg, and 2.56 mg). Afterwards, 3.4 μL of 2-mercaptoethanol was added and the mixture was kept for 10 min at 40 °C. Then, denaturated proteins were analyzed by SDS-PAGE and stained with Coomassie-blue [25].

### 4.8. Statistical Analysis

Regarding the cholera toxin-induced diarrhoea, the plot of percentage of inhibition against concentration was made; the best straight line was determined by regression analysis and the effective doses 50 (ED_50_) values were calculated. The regression coefficient, its level of significance (*p*), and correlation coefficient were calculated. The experiments were performed two times for each concentration. ED_50_ values are mean ± S.E.M. *p* < 0.05 (1-way ANOVA followed by Dunnett’s post-hoc test), GraphPad Prism Version 5.03 was used (GraphPad Software Inc., La Jolla, CA, USA).

Regarding the antibacterial assay, the lowest concentration of the sample in a plate that failed to show any visible macroscopic bacterial growth was considered as the MIC. The MIC determination was performed in duplicate, and the experiment was repeated two times.

## 5. Molecular Docking Analysis

### Molecular Docking of Incomptine A (IA), Incomptine B (IB), and Racecadotril (REC)

The incomptine A (**IA**), incomptine B (**IB**) and racecadotril (REC) structure was created and prepared using MOE software [26,27,28,29]. For conformational analyses, the first structure of Incomptine A was constructed using the Molecular Operating Environment (MOE. 2020), and charges were assigned with the MMFF94 forcefield and Gasteiger. Then, the obtained 3D model was subjected to a systematic conformational search with MOE software. The conditions of search were: rejection limit—100; iteration limit—10,000; RMS gradient—0.005; RMSD limit—0.25; MM iteration limit—500; strain cutoff—7; conformation limit—10,000. The generated conformation of lowest energy and most stable was analyzed and compared with the known three poses of incomptine A (retrieved from molecular docking study into site A and B).

The query molecules were screened against the three-dimensional crystal structure of the cholera toxin retrieved from the Protein Data Bank (CT, PDB 1XTC, R = 2.40 Å) using AutoDock Vina 1.1.2 with center coordinates of (x = 55.52, y = −20.99 and z = 1.077) for the active site from subunit A (CTA), then (x = −18.73, y = 9.812, and z = 5.146) for the primary binding site from subunit B (CTB), and (x = 2.049, y = −10.944 and z = −3.681) for the secondary binding site from subunit B (CTB). The dimensions of grids were 25 × 25 × 25 points with default spacing assigned by AutoDock Vina 1.1.2. [30,31]. The top 20 results were generated for each molecule and analyzed with PyMol. All water molecules and artifacts were removed from the crystallographic structures. Then, all polar hydrogen atoms were added, ionized in a basic environment (pH = 7.4) and Gasteiger charges were assigned.

## 6. Conclusions

In this study, we found that incomptine A (**IA**) possesses strong anti-diarrhoeal activity in cholera toxin-induced diarrhoea models in mice. The last result, together with the SDS-PAGE and in silico analysis results, suggests that **IA** has pharmacological potential as a therapeutic agent to combat acute secretory diarrhoea. Moreover, the different significant properties founded suggest that **IA** and **IB** may have complementary anti-diarrhoeal effects that explain the use of the leaves of *D. incompta* in Mexican traditional medicine to treat diarrhoea, the first as an antisecretory agent, and the second as an antibacterial compound. 

## Figures and Tables

**Figure 1 pharmaceuticals-15-00196-f001:**
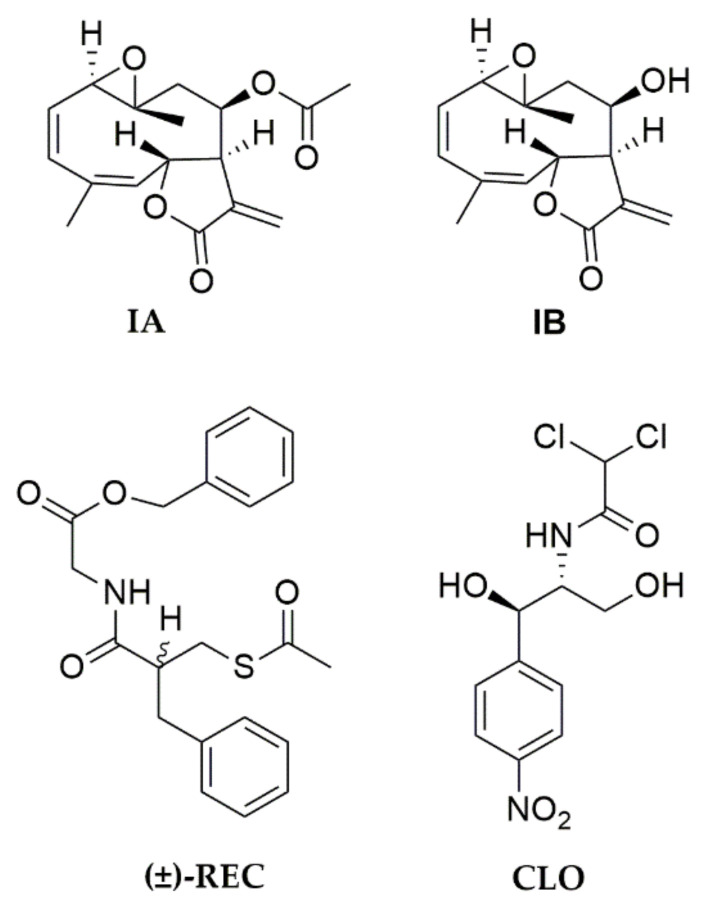
Structures of incomptine A (**IA**), incomptine B (**IB**), chloramphenicol (**CLO**), and racecadotril (**REC**).

**Figure 2 pharmaceuticals-15-00196-f002:**
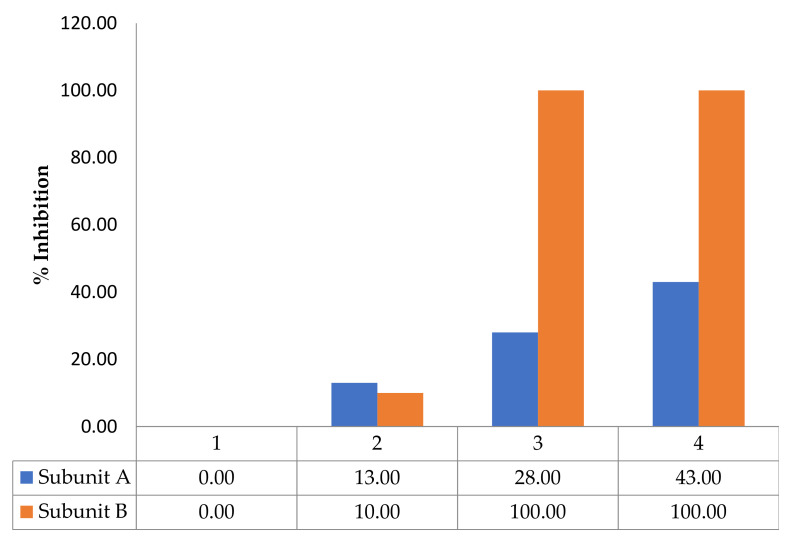
Binding properties of the incomptine A (**IA**) on cholera toxin (**CT**) analyzed by SDS-PAGE. Histograms of % of inhibition of subunits A and B from **CT** treated with **IA**. (1) **CT**, 10 μg (control); (2) **CT**: **IA**, 10 μg: 0.640 mg; (3) **CT**: **IA**, 10 μg: 1.28 mg; (4) **CT**: **IA**, 10 μg: 2.56 mg.

**Figure 3 pharmaceuticals-15-00196-f003:**
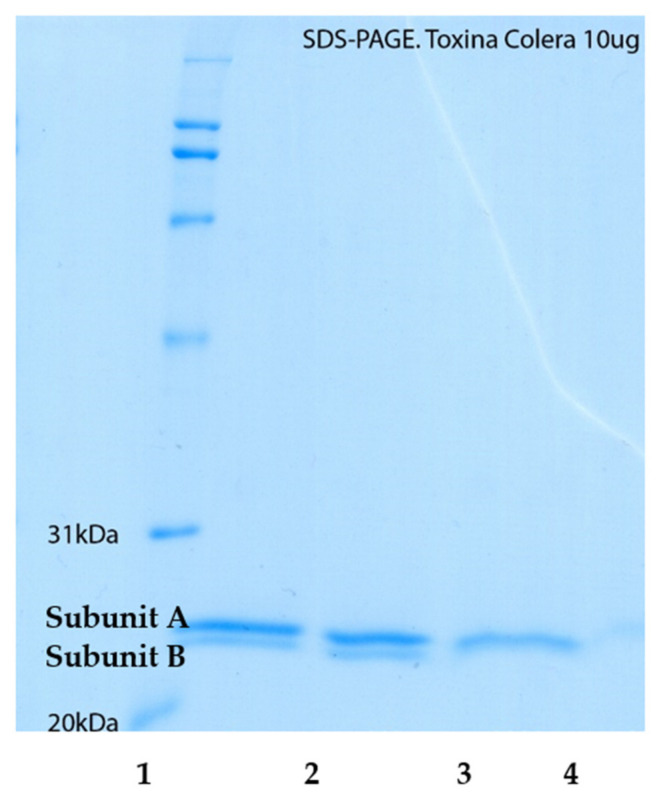
Binding properties of the incomptine A (**IA**) on cholera toxin (**CT**) analyzed by SDS-PAGE. SDS-PAGE stained with Coomassie-blue. (1) **CT**, 10 μg (control); (2) **CT**: **IA**, 10 μg: 0.640 mg; (3) **CT**: **IA**, 10 μg: 1.28 mg; (4) **CT**: **IA**, 10 μg: 2.56 mg.

**Figure 4 pharmaceuticals-15-00196-f004:**
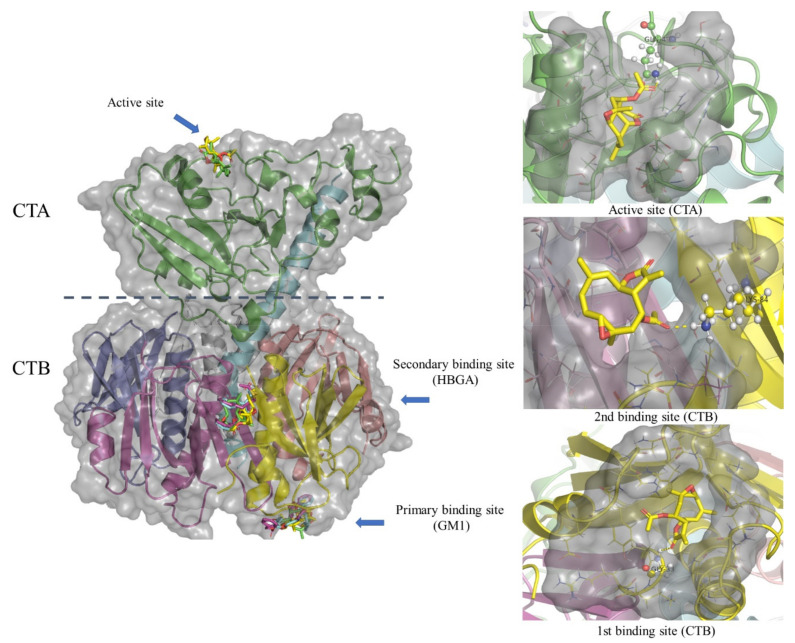
Schematic structure of cholera toxin (CT) and its A and B subunits. CT consists of one catalytic active site in subunit A (CTA: green color) and two binding sites in a pentamer subunit B (CTB: first and second binding site are colored in yellow and purple, respectively). Structures of incomptine A (yellow color), incomptine B (green color), R-racecadotril (purple color), and S-racecadotril (cyan color) are illustrated in all considered binding sites.

**Figure 5 pharmaceuticals-15-00196-f005:**
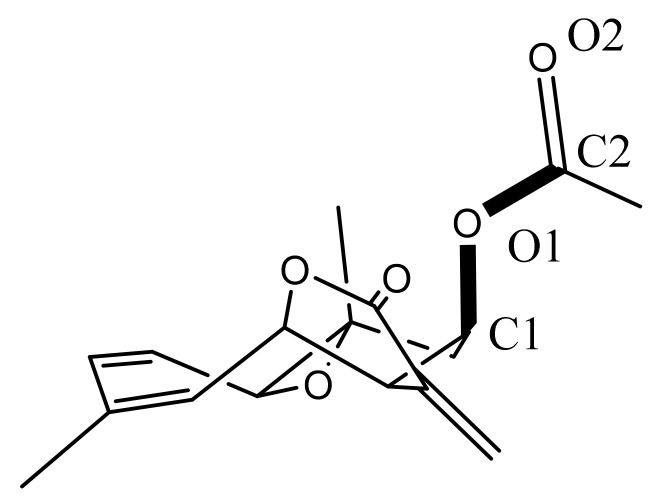
Rotatable bonds of Incomptine A.

**Figure 6 pharmaceuticals-15-00196-f006:**
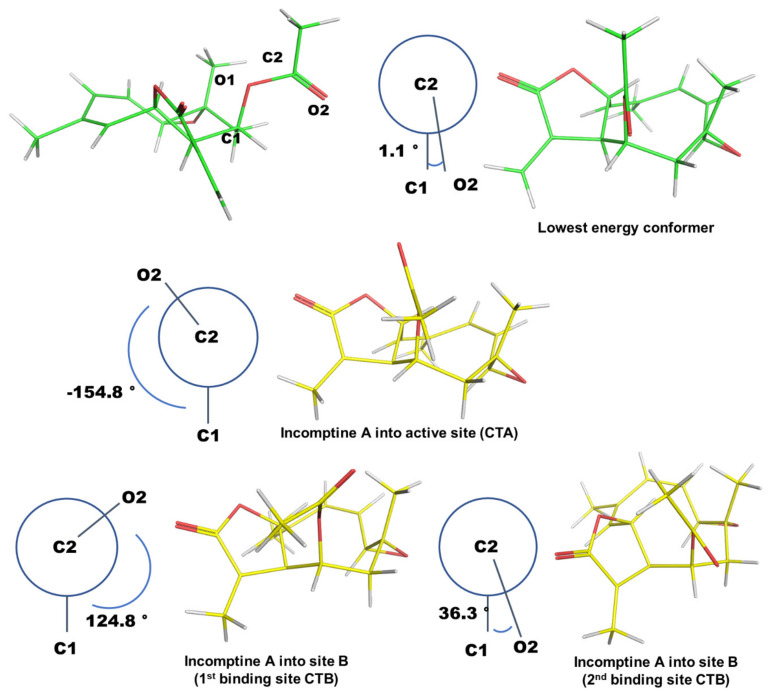
Dihedral angles of conformers of incomptine A (**IA**) and a comparison between all conformations (lowest energy conformer and three poses obtained from the binding sites of cholera toxin).

**Table 1 pharmaceuticals-15-00196-t001:** Antibacterial and anti-diarrhoeal activities of incomptines A (**IA**) and B (**IB**).

Treatment	*Vibrio cholerae*	Cholera Toxin-Induced Diarrhoea
	MIC (mg/mL) ^a^	ED_50_ (mg/kg) ^b^
Incomptine A (**IA**)	0.15	8.1 ± 1.3
Incomptine B (**IB**)	0.05	80.0 ± 3.7
Chloramphenicol (**CLO**)	> 2.0	-
Racecadotril (**REC**)		71.8 ± 2.7

^a^ Results are expressed as mean *(n* = 4). ^b^ Results are expressed as mean (*n*= 6) ± S.E.M., *p* < 0.05, Correlation coefficient > 0.9700.

**Table 2 pharmaceuticals-15-00196-t002:** Interactions of incomptine A (**IA**), incomptine B (**IB**), *R*-racecadotril (**REC**), and *S*-racecadotril with residues of binding sites of cholera toxin (CTA and CTB).

Compound	Subunit A (CTA)			Subunit B (CTA)					
	Active Site			Primary Binding Site (GM1)			Secondary Binding Site (HBGA)		
	ΔG ^a^	H-BR	NPI	ΔG	H-BR	NPI	ΔG	H-BR	NPI
**IA**	−5.9	Gln49	Ser61, Leu71, Thr75	−6.6	Gly33	Tyr12, His13, Asn14	−6.5	Lys84	Gln3, Asn4, Ile5
			Glu110, Glu112			Ala32, Glu51, Glu56			Phe25, Ser26, Thr28
						His57, Glu61, Trp88			Thr41, Ile47, Glu83
						Lys91			Ser100, Ala102
**IB**	−6.6	Gln49	Ser61, Leu71, Thr75	−6.1	--	Tyr12, His13, Asn14	−6.1	--	Gln3, Asn4, Ile5
		Arg54	Glu110, Glu112			Ala32, Glu51, Glu56			Phe25, Ser26, Thr28
						His57, Glu61, Trp88			Thr41, Ile47, Glu83
						Lys91			Ser100, Ala102
**R-REC**	−5.8	---	Ser61, Ser68, Leu71	−6.8	Glu61	Tyr12, His13, Asn14	−5.3	Thr41	Gln3, Asn4, Ile5
			Thr75, Asp109, Glu110		Trp88	Ala32, Glu51, Glu56			Phe25, Ser26, Thr28
			Glu112			His57, Ile58, Lys91			Ile47, Glu83, Ser100
									Ala102
**S-REC**	−5.7	--	Ser61, Ser68, Leu71	−7.0	His13	Tyr12, Asn14, Ala32	−5.4	Thrt41	Gln3, Asn4, Ile5
			Thr75, Asp109, Glu110		Trp88	Glu51, Glu56, His57			Phe25, Ser26, Thr28
			Glu112			Ile58, Gln61, Lys91			Ile47, Glu83, Ser100
									Ala102

^a^ ΔG: binding energy (kcal/mol); H-BR: H-bonding residues; NPI: nonpolar interactions; Asp: aspartate; Asn: asparagine; Arg: arginine; Gln: glutamine; Lys: lysine; Thr: threonine; Ser: serine; Trp: tryptophan; Leu: leucine; His: histidine; Gly: glycine; Glu: glutamic acid; Ile: isoleucine; Tyr: tyrosine; Phe: phenylalanine.

## Data Availability

Data sharing is not applicable to this article.

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
