# Peer review of "Understanding the Anti-Diarrhoeal Properties of Incomptines A and B: Antibacterial Activity against *Vibrio cholerae* and Its Enterotoxin Inhibition"

_pharmaceuticals, 2022, doi:10.3390/ph15020196_

Round 1

Reviewer 1 Report

Dear Author, 

The manuscript entitled "Understanding the anti-diarrhoeal properties of Incomptines A 2 and B: antibacterial activity against Vibrio cholera and its enter- 3 otoxin inhibition has been designed well. However, the comments need to be justified before considering into the journal. 

  1. The experimental design, how many groups have been used?
  2.  The discussion has potential to explore into more mechanistic way. 
  3. Any statistical analysis used in this study? 
  4. The references need to check for the uniformity and also the updated references. 

Author Response

Reviewer 1

Query1

  1. The experimental design, how many groups have been used?

Answer:

Currently there isn’t any specific strategy there are several approach’s including in vitro, in vivo, and in silico methods (see Komiazyk et al. BMC Complement Altern Med 2019,19,140; Heim et al. Sci Rep 2019, 9, 12243) all are complementary and depend of stages of V. cholerae infection (bacterial target, extracellular toxin target, human cell target, and in silico method on cholera toxin). In this contribution we reported: bacterial target, extracellular toxin target and in silico method on cholera toxin as target.     

Query 2

  1. The discussion has potential to explore into more mechanistic way

Answer:

In agreement with your suggestion the discussion was favored in the mechanism of action; lines 314-318.

Query3.

  1. Any statistical analysis used in this study?

Answer:

The statistical analysis section was included

Query 4

  1. The references need to check for the uniformity and also the update references.

Answer:

All references were checked for the uniformity and update

Reviewer 2 Report

Here, the authors aim at investigating the anti-Vibrio and antidiarrheal activities of two sesquiterpene lactones, incomptine A and incomptine B, in in-vitro, in-vivo and in-silico settings. However, in its current form, the present manuscript is very primitive and contains numerous technical errors as well as major omissions. Therefore, it cannot be accepted for publication in Pharmaceuticals. My main concerns are as follows:

1) CT binds to glycans on the surface of target cells through its B pentamer. This is followed by endocytosis and the release of the A1 portion of the protein complex (CTA1). CTA1 is responsible for ADP-ribosylating the human signaling protein G using NAD+. So what does "% inhibition of CT" mean in the present report? Did the authors assay the ADP-ribosylation activity of V. cholerae CT? Obviously, no.

2) V. cholerae CT is a multiprotein complex consisting of an A subunit (27 kDa) surrounded by five B subunits (12 kDa each). The authors' electropherogram lacks a ladder, making their subunit assignments quite suspicious and distrustful. And, more importantly perhaps, the disappearance of the bands for subunits A and B in the presence of incomptines does not make sense at all. Besides the fact that this does not correlate with an inhibition of the ADP-ribosylation activity of CT (as there are no G proteins and other minimum essential reagents in the reaction mixture), the loss of CT from the mixture is physically impossible. Did it just precipitate upon incomptine addition?

3) Given the fact that CT is a proteinaceous toxin, its intragastric administration to mice may result in complete proteolysis. Could not V. chloerae CT be intraluminally administered?

4) The data collected from the in-vivo experiments are missing. The authors are supposed to disclose the findings of the inferential statistical investigation with both control and experimental groups.

5) It is not correct to use V. cholerae CT in its inactive apo form for molecular docking purposes. A quick search of the Protein Data Bank (as of 19 December 2021) reveals that there exist only nine crystal structures for V. cholerae CT (UniProt ID: P01555) and that none of them is with a bound small-molecule inhibitor. So how did the authors identify active and allosteric sites on V. cholerae CT? In fact, using CTA1 in complex with its human protein activator ARF6 would be the best option in this case.

6) Racecadotril is an anti-secretory HUMAN ENKEPHALINASE INHIBITOR which is used in the treatment of diarrhea. So what is the rationale behind docking racecadotril onto V. cholerae CT and comparing the resulting docking scores (all of which appear to be relatively low) between racecadotril and incomptines?

7) Poor usage of English language is quite evident throughout the manuscript. (It is disappointing to see that even the taxonomic name of the bacterium of interest is wrongly spelt.)

Author Response

Reviewer 2

Dear reviewer thanks for your important and stimulant comments to improve our manuscript

Comments and Suggestions for Authors

Here, the authors aim at investigating the anti-Vibrio and antidiarrheal activities of two sesquiterpene lactones, incomptine A and incomptine B, in in-vitroin-vivo and in-silico settings. However, in its current form, the present manuscript is very primitive and contains numerous technical errors as well as major omissions. Therefore, it cannot be accepted for publication in Pharmaceuticals. My main concerns are as follows:

Comment 1

1) CT binds to glycans on the surface of target cells through its B pentamer. This is followed by endocytosis and the release of the A1 portion of the protein complex (CTA1). CTA1 is responsible for ADP-ribosylating the human signaling protein G using NAD+. So what does "% inhibition of CT" mean in the present report? Did the authors assay the ADP-ribosylation activity of V. cholerae CT? Obviously, no.

Answer:

Is true we don’t use AND-ribosylation assay. However, full inhibition of subunit A is clear in the SDS-PAGE assay a method used as an experimental design for other authors as cite Komiazyk et al. BMC Complement Altern Med 2019,19,140. Also, is clear the difference of effect on subunit B.  

2) V. cholerae CT is a multiprotein complex consisting of an A subunit (27 kDa) surrounded by five B subunits (12 kDa each). The authors' electropherogram lacks a ladder, making their subunit assignments quite suspicious and distrustful. And, more importantly perhaps, the disappearance of the bands for subunits A and B in the presence of incomptines does not make sense at all. Besides the fact that this does not correlate with an inhibition of the ADP-ribosylation activity of CT (as there are no G proteins and other minimum essential reagents in the reaction mixture), the loss of CT from the mixture is physically impossible. Did it just precipitate upon incomptine addition?

 Answer:

SDS-PAGE is an assay good characterized by our group to know the assignation of subunit A and B of cholera toxin (Velazquez et al. Journal of Ethnopharmacology. 103(1):66-70. 2006; Velazquez et al. Journal of Ethnopharmacology 126: 455-458. 2009; Velazquez et al. Journal of Ethnopharmacology Sep. 143(2): 716-719, 2012).  

3) Given the fact that CT is a proteinaceous toxin, its intragastric administration to mice may result in complete proteolysis. Could not V. chloerae CT be intraluminally administered?

Answer:

Also, cholera toxin-induced diarrhoea model in male Balb/c mice is validated for our group in this sense the intragastric administration of cholera toxin induces an important secretory diarrhoea (Pharmacognosy Magazine. 2017; 13 (50): 240-244) discarding proteolysis as your suggest .

4) The data collected from the in-vivo experiments are missing. The authors are supposed to disclose the findings of the inferential statistical investigation with both control and experimental groups.

Answer:

We include a Statistical section

5) It is not correct to use V. cholerae CT in its inactive apo form for molecular docking purposes. A quick search of the Protein Data Bank (as of 19 December 2021) reveals that there exist only nine crystal structures for V. cholerae CT (UniProt ID: P01555) and that none of them is with a bound small-molecule inhibitor. So how did the authors identify active and allosteric sites on V. cholerae CT? In fact, using CTA1 in complex with its human protein activator ARF6 would be the best option in this case.

Answer:

We used two different crystal structures of protein Data Bank and obtained positive result with secondary metabolites as flavonoids. Velazquez et al. Journal of Ethnopharmacology Sep. 143(2): 716-719, 2012.

6) Racecadotril is an anti-secretory HUMAN ENKEPHALINASE INHIBITOR which is used in the treatment of diarrhea. So what is the rationale behind docking racecadotril onto V. cholerae CT and comparing the resulting docking scores (all of which appear to be relatively low) between racecadotril and incomptines?

Answer:

Our result showed affinity of REC on cholera toxin (subunits) why validate its use in this experiment. Also, the affinity of incomptines was best or close or less compared with REC suggesting potential antidiarrheal properties for the SLs tested.

7) Poor usage of English language is quite evident throughout the manuscript. (It is disappointing to see that even the taxonomic name of the bacterium of interest is wrongly spelt.)

Answer:

All manuscript was checked

Reviewer 3 Report

Comments 1. The correct name is "Vibrio cholerae" line 27. 2.

The discussion of the results in table 2 needs to be redone, section 2.3 (line 195).

2.1 In table 2 regarding:

- subunit A (CTA), IB obtained better binding energy value compared to IA, R-REC and S-REC;

- subunit B (GM1), R-REC and S-REC obtained better binding energy values compared to IA and IB;

- B subunit (HBGA), IA obtained better binding energy value compared to other ligands.

2.2 IA obtained better results in two binding sites (GM1 and HBGA) compared to IB, which obtained better results in the CTA site. 

Author Response

Reviewer 3

Query 1

Comments 1. The correct name is “Vibrio cholerae” line 272

Answer:

All manuscript was checked

Query 2

The discussion of the results in table 2 needs to be redone, section 2.3 (line 195)

2.1 In table 2 regarding:

-subunit A (CTA), IB obtained better binding energy valued compared to IA, R-REC and S-REC;

-subunit B (GM1), R-REC and S-REC obtained better binding energy values compared to IA and IB.

- B-subunit (HBGA), IA obtained better binding energy values comparted to other ligands.

2.2. IA obtained better results in two binding sites (GM1 and HBGA) compared to IB, which obtained better results in the CTA site.

Answer:

Results in the section 2.3 and discussion were redone in agreement with table 2 and Figure 2.

Reviewer 4 Report

Manuscript pharmaceticals-1526680 presents the in-vitro screening of incomptines A and B against Vibrio cholerae, a diarrhea-causing bacterium; an in-vitro analysis of the interactions of the compounds with cholera toxin; an in-vivo investigation of the anti-diarrheal activity of the compounds in a mouse model; and an in-silico analysis of the molecular interactions of the ligands with cholera toxin using molecular docking.

Although not confusing to read, the manuscript would benefit by a careful proofreading by a native speaker of English.
For example, on line 204, replace "...was major than than IB." with "...was greater than that of IB."

The two incomptine ligands are medium-sized rings with several conformational possibilities that are not likely to be interconvertible.  The authors should carry out a conformational analysis of the three-dimensional structures to insure that the lowest-energy conformation is used for the molecular docking investigation.

Section 5.1., lines 415-416: More details are necessary for the preparation of the incomptines.  Was a conformational analysis carried out?  If so, at what level of theory?  What are the energy differences between the possible conformers?

Author Response

Reviewer 4

Manuscript pharmaceticals-1526680 presents the in-vitro screening of incomptines A and B against Vibrio cholerae, a diarrhea-causing bacterium; an in-vitro analysis of the interactions of the compounds with cholera toxin; an in-vivo investigation of the anti-diarrheal activity of the compounds in a mouse model; and an in-silico analysis of the molecular interactions of the ligands with cholera toxin using molecular docking.

Query 1

Although not confusing to read, the manuscript would benefit by a careful proofreading by a native speaker of English.

For example, on line 204, replace "...was major than than IB." with "...was greater than that of IB."

Answer:

The manuscript was checked by all authors. The mistake was corrected

Query2

The two incomptine ligands are medium-sized rings with several conformational possibilities that are not likely to be interconvertible.  The authors should carry out a conformational analysis of the three-dimensional structures to insure that the lowest-energy conformation is used for the molecular docking investigation.

Section 5.1., lines 415-416: More details are necessary for the preparation of the incomptines.  Was a conformational analysis carried out?  If so, at what level of theory?  What are the energy differences between the possible conformers?

Answer:

Incomptine A was minimized with calculations involving MM mechanics (MMFF94) to obtain a stable conformer, called local energy minimum.

Usually, in molecular docking, the protein is considered rigid, with the exception of ligands, which have certain flexibility during the process and they lose the global energy minimal conformation. Obviously, these considerations affect the quality of the generated poses, because, in reality, both protein and ligands have an induced fit. However, the focus of our work was to perform the analysis of the interactions between the protein and the ligand into the sites reported, and the purposes were identified the active pose through the ranking of different poses based on their scoring functions and not the global energy. To identify the molecular behavior, it is necessary the minimization of energy of 3d structures, but the level of analysis depends on the goal. Our objective was to study the site described on the protein and the interactions.

Although the most active conformer seems to be biologically potent, studies have shown that the bioactive conformed might differ from it. Therefore, the search strategy of ligand conformation is not useful and these methods are computationally too expensive for docking purposes.

References:

  • Pantsar T, Poso A. Binding Affinity via Docking: Fact and Fiction. Molecules. 2018 Jul 30;23(8):1899. doi: 10.3390/molecules23081899. PMID: 30061498; PMCID: PMC6222344.
  • Chapter 5 - Computational Chemistry Kunal Roy Supratik Kar Rudra Narayan Das, 2015.Pages 151-189
  • Guillaume Bouvier, Nathalie Evrard-Todeschi, Jean-Pierre Girault, Gildas Bertho, Automatic clustering of docking poses in virtual screening process using self-organizing map, Bioinformatics, Volume 26, Issue 1, 1 January 2010, Pages 53–60, https://doi.org/10.1093/bioinformatics/btp623

Round 2

Reviewer 1 Report

The manuscript may be accepted as the corrections have been made sufficiently. 

Author Response

Comments and Suggestions for Authors

Reviewer 1

The manuscript may be accepted as the corrections have been made sufficiently.

Answer: 

Queries weren’t required

Reviewer 2 Report

1) Komiazyk et al. (DOI: 10.1186/s12906-019-2540-6) state that "cholera toxin is separated into a 57 kDa pentameric subunit B (B5), a 28 kDa subunit A and a small amount of 11.4 kDa monomer subunit B". In the authors' electropherogram (Fig. 7), there exist two bands that migrate very close to each other. Are they the 28-kDa and 11.4-kDa bands? It is quite likely that the lower band could be a degradation product of the upper band. Intriguingly, subunit A appears to be greater in quantity than subunit B. The authors must include the entire electropherogram (not only a small cutout section of it) in their manuscript, along with a relevant protein ladder, to prevent bias. I have already asked this in my previous revision.

2) Komiazyk et al. (DOI: 10.1186/s12906-019-2540-6) also state that "SDS-PAGE analysis reveals that agrimony, raspberry leaf, blackberry leaf and wild strawberry leaf extracts cause aggregation of the toxin". This is the mechanism I have already suggested in my previous revision. In the current Pharmaceuticals manusript, the authors fail to explain their SDS-PAGE findings (if any) in a logical and reasonable manner.

3) I was unable to find qualitative or quantitative data as a direct proof of diarrhea induction by intragastric V. cholerae CT administration in the report by your research group (DOI: 10.4103/0973-1296.204564).

4) It is now more complete with the "Statistical Analysis" section.

5) The report by your research group (DOI: 10.1016/j.jep.2012.07.039) must be the worst reference that could be given here. It is pretty biased and irreproducible as you hide essential information regarding target selection and docking execution (no PDB IDs, no docking scores, etc.). So, neither this report nor your response answers the critical questions I have asked in my previous revision.

6) Which one of your results specifically demonstrate that the human enkephalinase inhibitor racecadotril directly binds to V. cholerae CT?

7) Even the corrections (yellow highlights) need an English check.

Author Response

Reviewer 2

1) Komiazyk et al. (DOI: 10.1186/s12906-019-2540-6) state that "cholera toxin is separated into a 57 kDa pentameric subunit B (B5), a 28 kDa subunit A and a small amount of 11.4 kDa monomer subunit B". In the authors' electropherogram (Fig. 7), there exist two bands that migrate very close to each other. Are they the 28-kDa and 11.4-kDa bands? It is quite likely that the lower band could be a degradation product of the upper band. Intriguingly, subunit A appears to be greater in quantity than subunit B. The authors must include the entire electropherogram (not only a small cutout section of it) in their manuscript, along with a relevant protein ladder, to prevent bias. I have already asked this in my previous revision.

Answer:

The electropherogram was included as Figure 3, in the paper this was showed in part because it was brake. Here I show complete. I agree a possibility is that bands could be a degradation product. If editor consider this part could be deleted of paper.

2) Komiazyk et al. (DOI: 10.1186/s12906-019-2540-6) also state that "SDS-PAGE analysis reveals that agrimony, raspberry leaf, blackberry leaf and wild strawberry leaf extracts cause aggregation of the toxin". This is the mechanism I have already suggested in my previous revision. In the current Pharmaceuticals manusript, the authors fail to explain their SDS-PAGE findings (if any) in a logical and reasonable manner.

Answer:

Additional text was included “lines 337-340

in this model this observation indicate that the mode of toxin inhibition depends upon the concentration of applied SL similar effect was reported to resveratrol. Also, could be that incomptines A and B caused aggregation of cholera toxin. Additional studies are necessary to discard the potential effect of IB on cholera toxin [6].

3) I was unable to find qualitative or quantitative data as a direct proof of diarrhea induction by intragastric V. cholerae CT administration in the report by your research group (DOI: 10.4103/0973-1296.204564).

Answer:

Incomptine B and racecadotril may be consider inactive theirs ED50 >71 compared against Incomptine A (ED50 8.1 mg/kg). This is in agreement with high propulsive effect caused by cholera toxin in mice treated with incomptine B and racecadotril.

4) It is now more complete with the "Statistical Analysis" section.

Answer:

Thanks for your observation

5) The report by your research group (DOI: 10.1016/j.jep.2012.07.039) must be the worst reference that could be given here. It is pretty biased and irreproducible as you hide essential information regarding target selection and docking execution (no PDB IDs, no docking scores, etc.). So, neither this report nor your response answers the critical questions I have asked in my previous revision.

Answer:

This reference was not included as support in this manuscript.

6) Which one of your results specifically demonstrate that the human enkephalinase inhibitor racecadotril directly binds to V. cholerae CT?

Answer:

This isn’t the objective of this work however in silico analysis suggest that REC may have effect on cholera toxin additional experiments are necessary to support this activity. 

7) Even the corrections (yellow highlights) need an English check.

Answer:

The manuscript was checked by all authors. In addition, 3 reviewers considering that is acceptable.
